# Public Opinions on Stray Cats in China, Evidence from Social Media Data

**DOI:** 10.3390/ani13030457

**Published:** 2023-01-28

**Authors:** Jiaping Xu, Aiwu Jiang

**Affiliations:** Guangxi Key Laboratory of Forest Ecology and Conservation, College of Forestry, Guangxi University, Nanning 530004, China

**Keywords:** stray cats, public opinion, management policy, social media data, China

## Abstract

**Simple Summary:**

Stray cat management is often under public discussion because of different perceptions toward these animals. Accordingly, understanding public opinions on stray cats is important for devising management strategies. Using user-generated content (UGC) from Weibo (Chinese Tweet), this study analyzed public opinions on stray cats for five topics: (1) human-stray cat interactions, (2) stray cat-related welfare, (3) public nuisances, (4) ecological impacts, and (5) actions to be taken for stray cat management. Results indicated that people’s interactions with and attitudes to these animals were distinct. Management implications were proposed based on the detailed results.

**Abstract:**

The management of stray cats is often contentious because public perceptions about these animals are different. Using user-generated content from Weibo, this study investigated Chinese citizens’ opinions on stray cats on a large scale. Through the techniques of natural language processing, we obtained each Weibo post’s topics and sentiment propensity. The results showed that: (1) there were some irresponsible feeding behaviors among citizens; (2) public perceptions of the ecological impacts caused by stray cats were unlike; (3) the trap-neuter-return (TNR) method served high support in public discussion; (4) knowledge about stray cats’ ecological impacts was positively correlated with support for the lethal control methods in management. Based on these findings, we suggested that management policies should be dedicated to (1) communicating to the (potential) cat feeders about the negative aspects of irresponsible feeding behaviors; (2) raising “ecological awareness” campaigns for the public as well as highlighting the environmental impacts caused by stray cats; (3) understanding citizens’ perceptions toward different management scenarios and making decisions accordingly. In addition, this study also suggested that social media data can provide useful information about people’s opinions on wild animals and their management. Policies would benefit by taking this source of information into the decision-making process.

## 1. Introduction

Worldwide, stray cat (*Felis catus*) population management is an important issue [1]. Reported complaints regarding stray cats include public health and environmental sanitation [2,3], public nuisances [4], ecological impacts of cats killing birds, small mammals, and reptiles [5,6,7], as well as the animal welfare of the cats themselves [8,9]. Thus, interventions are needed to control stray cat populations and reduce the associated problems.

Stray cat population management is often controversial, likely due to cats’ dual role as pets or pests depending on the context. People’s preferences for management strategies are also various [10,11,12,13]. One common strategy for managing stray cat populations is the trap-neuter-return (TNR) method which involves the capture, sterilization, and return of the cats to the capture site. This method is considered a humane approach and has been promoted by cat advocacy and animal welfare groups. While conservationists argue that the TNR method is often ineffective in controlling stray cat populations or reducing the negative environmental impacts caused by these animals [14,15]. Another strategy for managing stray cat populations involves permanent removal (trap and euthanize). This lethal control method is regarded as ecologically effective, but it might be less acceptable to some social groups, including females, urban residents, and those who hold wildlife rights value orientations [11].

Successful stray cat population management requires public support because sometimes ecologically effective interventions could be socially unacceptable and might evoke public opposition. Decision-makers need to understand public opinions about stray cats and devise management strategies or educational campaigns accordingly [16]. Traditionally, researchers use surveys, questionnaires, and observations to investigate public opinions of wild animals and social acceptance of certain management strategies. These methods are often financial-costly, time-consuming, and under other limitations such as non-respondent bias or self-selection bias [17]. Thus, other data sources are needed to complement traditional data collection for a better understanding of public opinions on stray cats.

One open-source of public information is user-generated content (UGC) from social media platforms. Given the popularity of social media networks in recent years, a growing number of people have registered as social media users and keep sharing personal opinions online, making social media platforms a center of public information [18]. UGC is a form of content created and shared by users based on their experiences, opinions, ideas, or feedback. Through natural language processing techniques such as content analysis, UGC can be useful to understand public opinions of wild animals [19,20].

To our knowledge, so far, no study has used UGC to understand public opinions on stray cats in China. People’s interactions with and attitudes to stray cats are distinct among demographic groups [4,21]. Accordingly, Chinese citizens’ opinions of stray cats and preferences for management strategies can be diversified. Different voices last on social media platforms, but little research has addressed public opinions of stray cats using this data source. Given that the free-ranging cat population in China is large and may cause profound ecological impacts [5], understanding public opinions of these animals and devising appropriate management strategies are important. In summary, this study used UGC from Weibo (Chinese Tweet) to analyze public opinions of stray cats.

## 2. Materials and Methods

### 2.1. Data Collection

This study used UGC from Weibo for data analysis. Weibo (https://weibo.com/ accessed on 29 December 2022) is a popular social media platform in China, with 224 million daily active users and 511 million monthly active users as of September 2020 [22]. It is also one of the most important platforms for public discussion in China. Weibo provides a platform for users to post their opinions freely, and users can like, share, and comment on each other’s posts. Concerning human-stray cat relationships, related issues were widely discussed by Weibo users during the data collection period; over 400 posts were collected each day. Thus, UGC from Weibo provides a chance for understanding public opinions on stray cats on a broad scale.

From 1 January to 31 July 2022, we used the keyword “stray cat” as a query to retrieve Weibo posts via the application programming interface (API) (https://open.weibo.com/ accessed on 29 December 2022). A total number of 102,721 posts containing the keyword ‘stray cat’ were retrieved during data collection. For each post, the following information was extracted: username, timestamp, descriptive content, and the number of comments, likes, and shares by other users (see Table 1 for a sample).

### 2.2. Data Preprocess

To improve the representativeness of the retrieved posts, we removed the records with duplicated content, irrelevant information (e.g., web links) in the content, and content over 250 words or less than 15 words. Hereafter, 89,803 valid records were left. To analyze the descriptive contents, we first used the ‘re’ toolkit in python 3.7 to remove the special characters such as emojis. Then, the “jieba” package was used for Chinese text segmentation. Finally, we used the stop words dictionary to remove the words with little meaning.

Thereafter, we used the word2vec model to vectorize words. Word2vec is a widely used algorithm based on neural networks [23]. Using large amounts of unannotated plain text, word2vec models learn relationships between words automatically. Using the cleaned descriptive contents (89,803 records), we trained a fifty-dimensional word2vec model and used it for similar word extraction. The data processing flow is depicted in Figure 1.

### 2.3. Topic Identification of Contents

We manually evaluated 4490 records (5% of the total dataset) and found that public opinions on stray cats can be classified into the following topics: (1) human–stray cat interactions, (2) stray cat-related animal welfare, (3) public nuisances, (4) ecological impacts, and (5) actions to be taken for managing stray cat populations. Within each topic, several subcategories (subtopics) were determined for more detailed classifications. The determination of subcategories was first decided by the authors’ review of records. Then, we excluded subcategories that took small percentages of the data (e.g., stray cat-related policies) or might be hard to identify through keywords (e.g., stray cat-related social incidents). The exclusion of these subcategories was due to the difficulty of identifying a small subset of data and the cost of misclassifications. Finally, thirteen subcategories within five topics were included in topic identification (Table 2).

To identify the concerning subtopics within each record, thirteen lexicons (one for each subtopic) were developed for matching. Each lexicon was developed following three steps. For example, to develop a lexicon for matching the subtopic ‘Humans get close to cats’, words with the meanings of “pat/petting” were included in the lexicon as keywords. Secondly, using the word2vec model, words with similar meanings or high relevance of keywords were retrieved and added to the lexicon. In the third step, using the KM website (https://kmcha.com/ accessed on 15 September 2022), we searched for synonyms of keywords and expanded the lexicon. Subsequently, the lexicon was used for identifying this subtopic in the test dataset (490 manually coded records). Its performance was calculated using an accuracy score.

Identical steps were undertaken to develop other lexicons. The performance of lexicons in topic identification was acceptable (Table 2). Thus, the developed lexicons were used for topic identification of the total dataset. Note, each piece of content may involve multiple types of information and can be coded with more than one subtopic or none as well.

### 2.4. Sentiment Analysis

Baidu provides a natural language processing (NLP) platform (https://ai.baidu.com/ accessed on 15 August 2022) for sentiment analysis of Chinese content. Through the Baidu API for NLP, we obtained the positive sentiment score of each record. This score reflects the positive propensity of each given content, it ranges between 0 and 1, with a value close to 1 indicating very positive sentiment and a value close to 0 indicating very negative sentiment.

### 2.5. Statistical Analysis

To understand the co-occurrence pattern of each pair of subcategories in public discussion, we used the Pearson correlation coefficient (*r*) to analyze the degree of linear correlation between each pair of variables. We used the *p*-value < 0.05 to present statistical significance.

## 3. Results

### 3.1. Results of Topic Identification

Topic identification results revealed that the topic of human-stray cat interactions took the largest percentage in public discussion (25.5%), followed by the topic of actions to be taken for stray cat management (20.9%) (Figure 2a). Other topics, including stray cat-related animal welfare (7.5%), public nuisances (3.4%), and ecological impacts (1.1%), took a small percentage of the public discussion. The identification of subcategories revealed that subtopics “Humans get close to stray cats” (10.3%), “Humans give food to stray cats” (15.6%), and “Stray cat adoptions” (15.9%) were most widely discussed (Figure 2b).

In addition to sharing personal opinions and experiences directly, Weibo users also express their attitudes on topics through comments, likes, and shares of other posts. We also calculated the rate of discussion for each topic by summing the total number of comments, likes, and shares per topic divided by the sum of the total number of posts per topic. Results of the discussion rate revealed that the topic of ecological impacts (90.7) had the highest discussion rate, followed by public nuisances (49.8) (Figure 2c). The discussion rates of subcategories showed that subtopics “To take the TNR method for stray cat population management” (111.4), “To take lethal control method for stray cat population management” (49.1), and “Stray cats making noise” (64) were highly discussed (Figure 2d).

### 3.2. The Correlations between Subtopics

The correlation matrix revealed the co-occurrence pattern of each pair of subcategories in public discussion. The correlation analysis results indicated that the subtopic “Stray cat adoptions” was positively correlated with “Care for stray cats” (r = 0.15) and “Humans get close to stray cats” (r = 0.16), but negatively correlated with “Humans give food to stray cats” (r = −0.07) (We only present correlations with |r| ≥ 0.05 in text, see Figure 3 for a complete correlation matrix). The subtopic “Care for stray cats” was positively correlated with “Humans get close to cats” (r = 0.05) and “Cats’ physical body conditions” (r = 0.05), but negatively correlated with “Humans give food to stray cats” (r = −0.06). The subtopic “To take lethal control methods” was positively correlated with “The ecological impacts” (r = 0.08).

The subtopic “Cats attack humans” was positively related to “Humans get close to cats” (r = 0.05). The subtopic “Humans give food to cats” was positively related to “Cats’ living conditions” (r = 0.07).

### 3.3. Results of Sentiment Analysis

The sentiment analysis results revealed that in most subtopics, public emotions tended to be negative (Figure 4). The medians of sentiment scores showed that public emotions tended to be positive to four subtopics: “Humans get close to cats” (median = 0.75), “Care for stray cats” (median = 0.79), “Stray cat adoptions” (median = 0.82), and “To take the TNR method for stray cat management” (median = 0.87). Six subtopics including “Cats attack humans” (median = 0.09), “Public sanitation” (median = 0.02), “Public health” (median = 0.2), “Cats making noise” (median = 0.15), “The ecological impacts” (median = 0.14), and “To take lethal control methods for stray cat population management” (median = 0.06) showed median sentiment scores less than 0.25, indicating public emotions towards these topics tended to be negative.

## 4. Discussion

The management of stray cats is often contentious because public perceptions about these animals are distinct [10,24,25]. To date, the majority of public opinion research has been conducted on questionnaire surveys and is often restricted to local scales. Large-scale data are still lacking. Using the social media data from Weibo, this study investigated Chinese citizens’ opinions on stray cats on a large scale. Our results revealed public opinions on stray cats in five topics. The percentage, discussion rate, and emotions toward each subtopic were analyzed to provide decision support for stray cat management.

### 4.1. Human-Stray Cat Interactions: Interventions Are Needed to Curb Irresponsible Feeding Behaviors

The percentage result showed that many Chinese citizens closely interact with stray cats. Within these interactions, feeding stray cats was the commonest. The correlation matrix showed that “Humans give food to stray cats” was negatively correlated with “Call for care for stray cats” and “Stray cat adoptions”. This result indicated that people who give food to stray cats were not likely to help these animals further (e.g., help find adoptions). This could reflect some irresponsible feeding behaviors among the citizens.

Feeding by humans reduces the average range size of free-ranging cats but increases the densities of cats in locations [26]. High densities of stray cats could increase the prevalence of parasites in the environment [2,3], augment the predation impacts on local biodiversity, as well as introduce public nuisances (e.g., property damage, cats making noise during mating). In addition, feeding stray cats may also introduce a legal liability for cat feeders if the actual breeding relationship or management responsibility is confirmed. As the Chinese Civil Law Code states “If any animal kept causes damage to another person, the animal breeder or manager shall be liable for a tort; However, if … liability may be exempted or mitigated.” Thus, feeding stray cats might put feeders into disputes or even lawsuits when the fed animals cause damage to another person.

Our review of the posts suggested that many cat feeders regarded feeding stray cats as a kindness and practiced it because of compassion. This feeding belief was also reported among Malaysian citizens [7]. In contrast, the result from sentiment analysis suggested that over half of the posts showed negative attitudes toward feeding stray cats. This result indicated that Chinese citizens’ beliefs about feeding stray cats were different [27,28]. Thus, before educational campaigns could be implemented for the target audience, future research should be designed to understand Chinese citizens’ beliefs about feeding stray cats, as well as to investigate the demographic characteristics of Chinese cat feeders. Communicating to the (potential) cat feeders about the negative aspects of irresponsible feeding behaviors may help mitigate this problem efficiently.

### 4.2. Ecological Impacts of Stray Cats: Disputes between Cat Advocates and Environment Conservationists

In public discussion, the ecological impacts of stray cats took the smallest percentage but had the highest discussion rate. Through manual evaluation, this result not only revealed people’s concern for the ecosystem but also involved the debates between cat advocates and environment conservationists. For example, some cat advocates insist that cats are a part of the ecosystem; overfed cats will not hunt wildlife, and that the ecological impacts of cats are minor compared with other anthropogenic factors. In fact, domestic cats have been listed among the 100 worst non-native invasive species in the world [29], and free-ranging cats have substantial impacts on local biodiversity [5,6,7,30], even overfeeding does not stop natural feline hunting behavior [31].

The disputes about the ecological impacts caused by stray cats seemed to reflect a deficiency in knowledge of cat behavior and ecology among some social groups. However, former research suggested that the majority of cat owners disagreed that free-ranging cats were a problem or harmful to wildlife despite being broadly aware of their cats’ predatory behavior [32]. The author explained that cat owners’ favorable view of owned property (i.e., pets) might result in this distorted assessment of their cats’ predatory habits [33]. In the current study, cat advocates’ neglect of stray cats’ ecological impacts could also result from their affection for cats. If this is the case, focus on “ecological awareness” campaigns appear unlikely to influence cat advocates’ support for certain management options (e.g., trap and euthanasia). Instead, better motivation to accept controls on free-ranging cats may be achieved by highlighting welfare advantages [34].

### 4.3. The TNR and Lethal Control Methods as Management Strategies

In public discussion, posts about specific management strategies (i.e., the lethal control method and the TNR method) took a small percentage but had a high discussion rate. The TNR method has been considered a humane treatment of stray cat populations and received high support in public discussion. As a comparison, the lethal control methods received more opposition in the discussion. These findings are also supported by the sentiment analysis results, with public emotions being positive toward the TNR method and negative toward the lethal control methods. A similar preference for the TNR method was also reported in other countries [10,11,13].

This preference for stray cat population management is explained as a shift in ethics over time, reflected by a growing consensus that cats (and other non-human animals) also have intrinsic value and deserve to be treated with compassion [35]. However, cumulative evidence suggested that the TNR method was often ineffective in decreasing stray cat populations [15]. For example, long-lasting TNR programs in Rome saw a 16–32% decrease in urban feral cat population size but concluded that TNR programs alone are not sufficient for managing the urban feral cat population if abandonment of cats is not stopped [36]. Modeling indicates a high sterilization rate (e.g., 71–94% of cats) is necessary for TNR to initiate consistent declines in cat population sizes [37]. Although the number of stray cats in China is unclear, an estimation of owned cat population indicated that over 34 million owned-cat were free-ranging in Chinese urban areas, and this number exceeds 60 million in rural areas [5]. Thus, for the TNR method to be continuously implemented, financial support from non-government organizations or the citizens will be needed due to the large population size of cats in China. Moreover, an increase in monitoring and regulatory oversight of the TNR method is needed for the well-being of both cats and other wildlife [7].

“To take lethal control methods for stray cat population management” was positively correlated with “ecological impacts”. This result indicated that public knowledge of the negative ecological impacts caused by stray cats may influence their preference for management strategies. Future research could benefit by taking this factor as an explanatory variable for understanding public preference for stray cats (or other invasive animal species) management. In addition, communicating to the public about the negative ecological impacts caused by cats may enlist popular support for ecologically effective management strategies (e.g., lethal control methods). Since the TNR method has restrictions in practice, alternative and/or complementary approaches are needed. Communicating to the public about the necessity of these approaches may help reduce opposition.

Former research has also suggested that the public widely supports forming responsible ownership and reducing cat abandonment [10,38]. We did not identify this subtopic because of the small percentage of related records. However, in post reviews, we found some posts strongly recommend fostering responsible ownership through legislation. So far, we only identified two general strategies (the lethal control method and the TNR method) in stray cat population management and thoroughly investigated public perceptions. A former study suggested that management strategies for the stray cat population can be classified into seven scenarios: household cat neutering with financial support for the owner, household cat neutering without financial support for the owner, encouraging responsible household cat ownership, trapping stray cats and taking them to a shelter, trapping and neutering stray cats for release into a managed “cat colony”, trapping and killing of stray cats, and undertaking no action [10]. Future research might benefit decision-making by investigating Chinese people’s preference for management strategies and the related factors in detail.

### 4.4. Limitations

The current study investigated Chinese people’s opinions on stray cats using social media data but it also had a few limitations for generalization. First, the Weibo data center suggested that most Weibo users (78%) aged below 30 and more than half of the users (54.6%) were females [22]. Previous research also suggested that females were more concerned about animal welfare and tended to respond to cat-related surveys (e.g., [10,38]). Besides, people are more likely to share joyful moments or post things that will obtain likes and be acceptable [39,40]. Thus, this study might report an incomplete representation of the Chinese population, as well as the biased positive sharing of personal experiences and perceptions of stray cats. Future research should benefit by correcting the demographic variables to a population level to see the differences in public opinions. Second, we analyzed text content only. Other data sources such as photos could also provide information for understanding human-stray cat relationships. Thus, future research could benefit from analyzing multiple information sources. Third, in this study, we considered stray cats as non-owned cats that range freely. However, sometimes it may be difficult to identify if a free-ranging cat is owned property or not because a number of Chinese cat owners allow their cats to range outside [5]. Due to this misidentification, posts about stray cats could also include people’s opinions on some owned free-ranging cats. Last, the Pearson correlation coefficients in this study were relatively low because the collected data were a random sample. Further pre-designed studies are needed to confirm the relationships between topics.

## 5. Conclusions

This study analyzed Chinese citizens’ opinions on stray cats using social media data. Our results indicated that (1) there were some irresponsible feeding behaviors among citizens; (2) public perceptions of the ecological impacts caused by stray cats were unlike; (3) the TNR method served high support in public discussion; (4) knowledge about stray cats’ ecology and behaviors were positively correlated with support for the lethal control methods in management. Based on these findings, we suggested that management policies should be dedicated to (1) communicating to the (potential) cat feeders about the negative aspects of irresponsible feeding behaviors; (2) raising “ecological awareness” campaigns to the public as well as highlighting the environmental impacts caused by stray cats; (3) understanding citizens’ perceptions toward different management scenarios better and making decisions accordingly.

## Figures and Tables

**Figure 1 animals-13-00457-f001:**
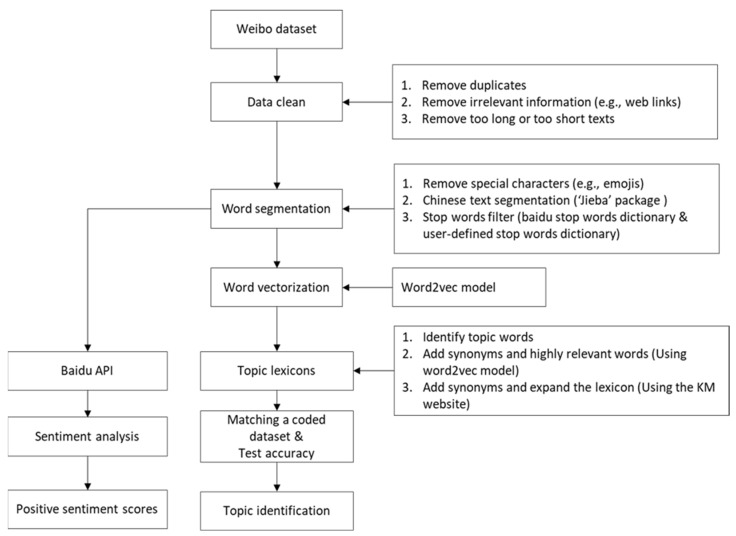
The flow of data processing.

**Figure 2 animals-13-00457-f002:**
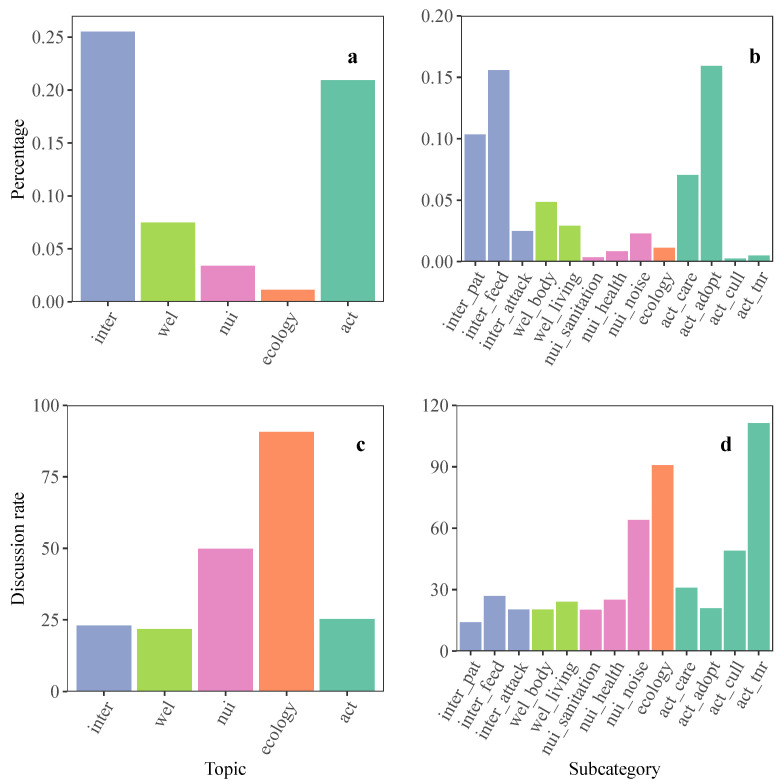
The percentage and discussion rate of identified topics. (**a**) The topics of human-stray cat interactions and actions to be taken for stray cat management took the largest percentage in public discussion. (**b**)The subcategories “Stray cat adoptions”, “Humans get close to stray cats”, and “Humans give food to stray cats” took the largest percentage in public discussion. (**c**) The topics “ecological impacts” and “public nuisances” had the highest discussion rate. (**d**) The subcategories “To take the TNR method for managing stray cat population”, “The ecological impacts”, and “Stray cats making noise” had the highest discussion rate.

**Figure 3 animals-13-00457-f003:**
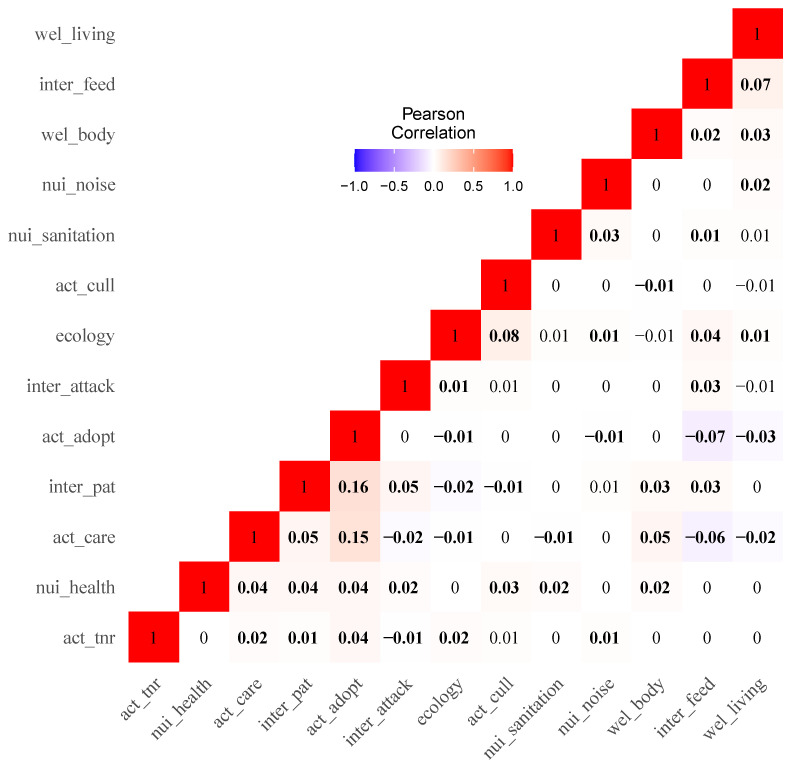
The correlation matrix of thirteen subtopics (Bold font highlights correlations with statistical significance, *p* < 0.05).

**Figure 4 animals-13-00457-f004:**
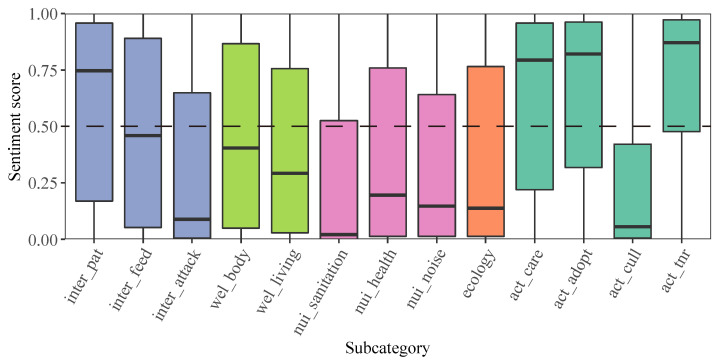
The sentiment scores of thirteen subtopics.

**Table 1 animals-13-00457-t001:** A sample of collected information from one post. The username was masked for privacy protection. The original information was Chinese and translated into English here.

Username	Date	Descriptive Content	Share	Comment	Like
***	12 July 2022	Care for animals around you. You may not love them, but please don’t hurt them.	0	0	3

**Table 2 animals-13-00457-t002:** Public opinions on stray cats can be classified into five topics. The performance of lexicons in topic identification was presented in accuracy scores. A portion of topic words (keywords within each subcategory) was presented.

Topic	Subcategory	Abbr.	Topic Words	Accuracy
Human-stray cat interactions	Humans get close to cats	Inter_pat	Petting; Pat	0.978
	Humans give food to cats	Inter_feed	Feed	0.987
	Cats attack humans	Inter_attack	Scratch	0.991
Animal welfare	Cats’ physical body condition	Wel_body	Health; Disabled; Sick	0.978
	Cats’ living condition (e.g., food resource/shelter)	Wel_living	Lack of food; Hungry	0.987
Public nuisances	Public sanitation	Nui_sanitation	Cleanness; Feces	1.000
	Public health	Nui_health	Parasite; Flea; Infectious	0.998
	Cats making noise	Nui_noise	Cry of the cat; Noise	0.991
Ecological impacts	Ecological impacts	Ecology	Bird; Ecosystem	0.998
Actions to be taken	Call for care for stray cats	Act_care	Care; Help	0.993
	Stray cat adoptions	Act_adopt	Adoption	0.971
	Lethal control methods	Act_cull	Cull; Euthanasia	1.000
	The TNR method	Act_tnr	Trap; Neutral; Return	1.000

## Data Availability

Third party data.

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
