# Peer review of "Public Opinions on Stray Cats in China, Evidence from Social Media Data"

_animals, 2023, doi:10.3390/ani13030457_

Round 1

Reviewer 1 Report

Comments on Manuscript "Public Opinions on Stray Cats in China, Evidence from Social Media Data" Submitted to the Animals

General Comments

I appreciate the opportunity to review this interesting manuscript. The study looks at the attitudes and comments of users of the Chinese network Weibo towards stray cats. By applying discourse analysis methods and developing an algorithm to select responses to the most relevant topics, the authors arrived at thought-provoking results. The highlights of the results were: a) the contradictory aspects between cat feeders and opponents; b) increase awareness campaigns on the environmental impact of stray cats; c) understanding how people perceive dealing with stray cats. In addition, the study confirmed the importance of social media for studying people's opinion trends, which is useful for defining social and environmental policies, etc.

The manuscript is well-written, allowing for fluent reading and good understanding.

The introduction is convincingly justified and well-founded in the scientific literature. Likewise, the methods are detailed and, with the necessary adjustments, it seems to allow the study to be reproduced on other social networks, in other languages, and with other subjects. Therefore, the heuristic value of the method is a strong part of the study.

In the results, the authors suggest that the p<0.05 correlations are important for the co-occurrence of the patterns of each pair of subcategories in the public discussion. The authors' interpretation of Weibo users' expressions in relation to stray cats relies heavily on these correlations. What is remarkable, however, is that the correlation values (positive or negative) are low. In other studies, the low values of r in the Pearson correlation test are usually negligible, even when there is alpha significance. I propose that the authors explain the statistical rationale for assuming such low correlations for the development of subtopics and their interrelationships. If the decision to assume low correlations is idiosyncratic, I would suggest adding some weaknesses of the study to the text. Incidentally, the description of the weaknesses of the study is well described.

Aside from that statistical query above, I found the results and discussion interesting. For example, the high percentage of people on the Weibo network who are interested in environmental pollution, how we treat cats, and the annoyance cats cause is important. In general, there seems to be a polarization regarding the socio-ecological role of stray cats in China.

The authors call for more user education about the behaviour and ecological role of cats. It is unknown how to deal with this problem, which is widespread in China. These results do not appear to differ significantly from studies in other culturally very different countries. I suggest that cats' roles are more unifying than traits that distinguish different cultures in other countries.

Surprisingly, people who use Weibo do not seem concerned about cats' public health role as carriers, vectors, and hosts of pathogens. Do the authors have any explanation why Weibo users, and of course the Chinese, do not discuss public health problems related to stray cats?

Despite my comments above, I do not think the discussion is contradictory or weak. On the contrary, the discussion prudently sticks to the results without ceasing to provoke further research into stray cats.

Citations in the text are appropriate and important clues for the argumentation and development of the text.

In general the manuscript is very good.

Author Response

We appreciated your recommendations for this manuscript. All suggestions were considered and adaptations were made for improvement.  Please see the details in the attachment.

Reviewer 2 Report

I like your paper. My comments are minor:

Line 21 and across the whole text: You intensively use the word “diversified,” which is inappropriate in most cases, specifically when you use this regarding public perceptions. The verb “diversified” means “made more diverse or varied.” Instead, you can use adjectives “different, district, unlike, mismatched,” and many others.

Line 35: I recommend replacing the word “involving” with “regarding” and “concerns.”

Line 43: Place the abbreviature TNR in parenthesis after “trap-neuter-return).

General comment: Unfortunately, you didn’t ask questions related to knowledge of cat fleas and cat-associated infections such as cat scratch disease, toxoplasmosis, and others. If you continue your research in this direction, I recommend thinking about that. Meanwhile, you can acknowledge the shortage in the section “Limitations.”  

Reviewer 3 Report

This manuscript presents an approach to public opinion regarding feral cat population management. I believe the topic is relevant from a conservation and One Health perspective and the use of social media to analyse public opinion is a good method to achieve the study purposes. Therefore, I think authors should receive credit for their work.

I only have some corrections/suggestions to the authors that I believe would improve their work significantly.

1) L13-14 "Results indicated that people’s interactions with and attitudes to these animals were distinct." - distinct how? Please explain briefly.

2) I suggest the authors also give a One Health approach to their work. Feral cat populations often carry diseases (Toxoplasmosis, FIV, FelV...) that may affect people and the cats they have in their homes. Thus, besides a conservation and ecological problem (as the authors mention in their introduction) it may also affect different living being for this reason. This will provide even broader importance to your manuscript.

3) Authors should end their Introduction section by presenting the aims of the study in an isolated last paragraph.

4) Authors should provide the statistical tests and significant p-value you considered at the end of the Methodology section (0.05).

5) L319 "Chinses", please correct this.

6) I do not understand what the small numbers (like "4" in line 36) mean. I was expecting brief explanations of some concepts, but I did not find them. Then, I thought it may be reference numbers in another format rather than "[4]". Please confirm this aspect and revise the whole manuscript because there are more cases.
